# Community-intrinsic properties enhance keratin degradation from bacterial consortia

Poonam Nasipuri[1], Jakob Herschend[1], Asker D. Brejnrod[2], Jonas S. Madsen[1], Roall Espersen[3], Birte Svensson[3], Mette Burmølle[1], Samuel Jacquiod[1¤‡], Søren J. Sørensen[1‡]*

1 Section for Microbiology, Department of Biology, University of Copenhagen, Copenhagen, Denmark, 2 Novo Nordic Foundation Center For Basic Metabolic Research, University of Copenhagen, Denmark, 3 DTU Bioengineering, Department of Biotechnology and Biomedicine, Technical University of Denmark, Lyngby, Denmark

☺ These authors contributed equally to this work.
¤ Current address: Agroécologie, INRA Centre Dijon, Université de Bourgogne Franche-Comté, Dijon, France.
‡ These authors also contributed equally to this work.
* sjs@bio.ku.dk

**Data Availability Statement:** Mass spectrometry data had been made available through ProteomeXchange Consortium via the PRIDE

## Abstract

Although organic matter may accumulate sometimes (e.g. lignocellulose in peat bog), most natural biodegradation processes are completed until full mineralization. Such transformations are often achieved by the concerted action of communities of interacting microbes, involving different species each performing specific tasks. These interactions can give rise to novel "community-intrinsic" properties, through e.g. activation of so-called "silent genetic pathways" or synergistic interplay between microbial activities and functions. Here we studied the microbial community-based degradation of keratin, a recalcitrant biological material, by four soil isolates, which have previously been shown to display synergistic interactions during biofilm formation; *Stenotrophomonas rhizophila*, *Xanthomonas retroflexus*, *Microbacterium oxydans* and *Paenibacillus amylolyticus*. We observed enhanced keratin weight loss in cultures with *X. retroflexus*, both in dual and four-species co-cultures, as compared to expected keratin degradation by *X. retroflexus* alone. Additional community intrinsic properties included accelerated keratin degradation rates and increased biofilm formation on keratin particles. Comparison of secretome profiles of *X. retroflexus* mono-cultures to co-cultures revealed that certain proteases (e.g. serine protease S08) were significantly more abundant in mono-cultures, whereas co-cultures had an increased abundance of proteins related to maintaining the redox environment, e.g. glutathione peroxidase. Hence, one of the mechanisms related to the community intrinsic properties, leading to enhanced degradation from co-cultures, might be related to a switch from sulfitolytic to proteolytic functions between mono- and co-cultures, respectively.

partner repository with the dataset identifier PXD016745.

**Funding:** The projected was funded by SJS as part the Keratin2Protein grant from The Danish Council for Strategic Research, The Programme Commission on Health, Food and Welfare. There was no additional external funding received for this study.

**Competing interests:** The authors have declared that no competing interests exist.

## Introduction

Utilization of microorganisms for industrial purposes is gaining momentum due to their associated economically and environmentally friendly benefits, contrasting current chemical or mechanical approaches. While biotechnological application of microorganisms has mostly focused on single strains, recent research highlights several advantages with mixed species communities. Mixed species communities are generally more robust when exposed to environmental changes and have been shown to outperform their single species constituents [1]. However, assembly, development, and function of such communities often fall short of expectations, as community-wide properties and characterization typically cannot be directly inferred from studying single species [2,3]. This phenomenon can be referred to as community-intrinsic properties; properties of a complex community that cannot be linearly extrapolated based on knowledge of the individual community members [3].

Microbial communities are partly governed by interspecies interactions, and understanding these relationships is key to artificially manipulate community assembly and shape desired properties. For biotechnological application, community-intrinsic properties hold a considerable potential as they may be synergistic, significantly enhancing productivity beyond the mere capacities of either individual species or the sum of single species [4]. Such community synergies have been shown in several cases e.g. with increased biofilm formation from mixed species communities [5–7] and also for degradation of complex recalcitrant organic molecules such as lignocellulose [8]. A variety of different factors can contribute to the emergence of intrinsic community properties, including substrate complexity that may lead to positive interactions among community members, enabling degradation of recalcitrant substrates [9] or yield complementary by-products [10]. Similarly, unique spatial distributions can facilitate direct or indirect cross-feeding mechanisms [11–13].

Here community-based degradation of keratin was compared to that of single species, to explore if the community-intrinsic property concept could potentiate this important process. The low-cost substrate keratin is biotechnologically interesting, as it can potentially be decomposed into high-value compounds through microbial conversion [14,15]. Keratin is a complex proteinaceous biomaterial with fibrous, insoluble and recalcitrant structures. Worldwide, keratin is a common by-product from livestock production, which accumulates in massive amounts [16]. In Denmark, according to the slaughterhouse Danish Crown, 17.5 million pigs are slaughtered each year, leading to a tremendous accumulation of keratin wastes. Keratins are classified as category 3 by EU parliament regulation [17], and improper disposal of this by-product can lead to critical pollution [18]. Keratin has potentially high nutritional value, as it contains 2–5% sulphur, 15–18% nitrogen, 3.2% mineral elements, 1.3% fat and 90% protein [16,19,20]. Decomposition products can therefore be made utilisable in the production of animal feed [18,21]. Nevertheless, conventional keratin treatment prior to incorporation into animal feed relies on expensive and thermo-chemically harsh procedures, yielding a poor product quality with non-digestible molecules [22–26]. Bioconversion stands as an interesting alternative for keratin valorisation, relying on enzymes produced by microorganisms (keratinases and other proteases). Indeed, application of enzymes in biotechnology is a commonly used alternative to chemicals because of their efficiency and cost-effectiveness [17,27–29]. Keratinase activity is widespread in the microbial world [14,30,31], and believed primarily to be associated with the S08 protease family [32] and characterized as serine endoproteases [30,33]. Recent reports suggest that keratinolytic activity is also found in proteases like metalloprotease [34–36], aspartic protease, and exoproteases like carboxyl protease, amino protease and dipeptidyl peptidase, [28,31,37,38]. Most research on keratinolytic bacteria (bacteria producing keratinase) has focused on various representatives from the genera *Bacillus* [39–41], *Chryseobacterium*

[35,42], *Streptomyces* [34], *Serratia* [43], *Lysobacter* [44], *Pseudomonas* [45], and specific species including; *Stenotrophomonas* sp. D-1 [46], *Microbacterium sp.* [47], and *Xanthomonas maltophilia* [48]. Although, application of single species can result in full keratin degradation, community-based features, such as e.g. enhanced or faster degradation through synergistic community interactions, make investigations of community application an interesting avenue for future research.

In the current work, we demonstrate enhanced keratin degradation as a community-intrinsic property. We utilise a well-characterised bacterial consortium consisting of *S. rhizophila*, *X. retroflexus*, *M. oxydans* and *P. amylolyticus*. These four strains have been co-isolated from soil [49] and have previously been shown to interact synergistically in biofilm formation in various settings [6,50]. *X. retroflexus*, was the best keratin degrader in terms of culture density and keratin degradation, and was here combined with the non-degraders *M. oxydans* and *P. amylolyticus*, and/or *S. rhizophila* that did degrade keratin but grew slowly, to investigate community degradation as compared to single species.

## Materials and methods

### Growth of bacterial isolates

The four bacterial strains used in this study were previously isolated from soil and identified by 16S rRNA gene sequence analysis [49,51]. *Stenotrophomonas rhizophila* (S), *Xanthomonas retroflexus* (X), *Microbacterium oxydans* (M), and *Paenibacillus amylolyticus* (P) were stored in 20% glycerol stocks and plated on 1.5% agar plates complemented with tryptic soy broth (TSB) (Sigma-Aldrich, 30 g/L) (TSA), followed by incubation for 48 hrs at 24˚C. Single colonies were picked from the TSA plates, and used to inoculate liquid cultures of TSB. Liquid cultures were incubated at 24˚C with horizontal shaking at 250 rpm for approx. 16 hrs, and used the following day to inoculate keratin liquid medium (KLM). KLM contained 0.5 g/L $NH_4Cl$, 0.5 g/L NaCl, 0.3 g/L $K_2HPO_4$, 0.4 g/L $KH_2PO_4$, 0,1g/L $MgCl_2.6H_2O$ and 10 g/L milled pig bristles and hooves (Lin et al.1992), and was adjusted to pH 7. The KLM cultures were grown in 250 mL shake flasks for 4 days at 200 rpm and 24˚C. Cultures contained 100 mL KLM and were inoculated with 1 mL single species culture adjusted to OD 0.7, or 1 mL of a mixed species culture assembled in equal proportions from OD 0.7 adjusted mono-cultures. A conversion table for OD to CFU has been appended to supplemental information, see S1 Fig and S1 Table. $OD_{600nm}$ was measured each day and viable cell counts were performed at day 4 by plate spreading on TSA, complemented with congo red (Sigma-Aldrich, 40 μg/mL) and coomassie brilliant blue G250 (Sigma, henceforth referred to as 'coomassie', 20 μg/mL). *M. oxydans* and *P. amylolyticus* could be selectively distinguished from *S. rhizophila* and *X. retroflexus* by their distinct colony morphology. A combined count was obtained for *S. rhizophila* and *X. retroflexus* from standard TSA plates with congo red and coomassie. Selective counts of *X. retroflexus* could be obtained by adding Kanamycin (50 μg/mL) to the TSA plates with congo red and coomassie.

All experiments included three biological replicates, each reproduced with three technical replicates.

### Protease, keratinase and protein concentration measurements

Protease activity was measured by degradation of azocasein (Sigma) as described by Jahan *et al.*, 2010 [52]. In summary, 200 μL of 1% azocasein solution (in 50 mM Tris-HCl buffer at pH 8.0) was incubated with 400 μl of supernatant in 24 well plates at 24˚C and 200 rpm. After 30 min. the reaction was stopped by addition of 1.4 mL 10% trichloroacetic acid (TCA), followed by 15 min. incubation at 4˚C. The solution was centrifuged at 10000g for 10 min. Thereafter, 1 mL of supernatant was mixed with 1 mL 0.5 M NaOH and the absorbance was

measured at 415 nm on a BioTek-ELx808 plate reader using the Gen 5.2.05 software. Keratinase activity was measured by degradation of azokeratin according to Riffel et al., 2003 [53]. The origin of the keratin and a detailed description of the azokeratin preparation is available from Kang et al., 2018 [15]. In summary, 800 μL of azokeratin solution (in 50 mM Tris-HCl buffer, pH 8.0) was incubated with 500 μL of the supernatant for 60 min. at 24°C and 200 rpm in a 24 well plate. The mixture was centrifuged at 10000g for 10 min. The supernatant was transferred to a 96 well plate and absorbance was measured at 415 nm.

The amount of enzyme required to increase the absorbance by 0.01 was defined as one unit (U) of enzyme activity under the given conditions. Protein concentration was measured with a Bradford assay (Bio-Rad) [54] according to manufactures description with bovine serum albumin as the standard.

## Keratin mass loss

After 4 days of growth, KLM bacterial cultures were filtered through pre-weighed 12–15 μm particle retention filter paper (Frisenette APS). The keratin in the culture broth was washed thoroughly, while on the filter paper, to remove planktonic and loosely attached bacteria. The filter paper was dried for 48 hrs at 50°C and re-weighed to calculate the keratin mass loss. Media without any bacteria added was used to normalize the keratin loss.

## DNA extraction from biofilm adhering to keratin particles

Day 2 and 4 cultures were filtered through 12–15 μm particle retention filter paper (Frisenette APS). The keratin particles in the culture broth were washed twice with 50 mL 0.9% NaCl solution, while on the filter, to eliminate the non-adherent planktonic cells. The filter paper was dried for 1 hrs at 50°C. Equal amounts (0.2 g) of keratin were recovered from each sample and DNA extraction was done using FastDNA SPIN Kit for Soil (MP Bio). The DNA samples were diluted 10 times and 2 μL of DNA template was used for qPCR analysis with Kapa sybr fast qPCR kit (Sigma-Aldrich). Each reaction of 20 μL contained; 10 μL of 2x Kapa Sybr Fast qPCR Master Mix, 6.8 μL of PCR-grade water, 0.4 μL of each forward (341F – 5'CCTACGGGAGGC AGCAG3') and reverse (518R – 5'ATTACCGCGGCTGG3') universal eubacterial 16s rDNA primers, 0.4 μL of 50X ROX High (Sigma-Aldrich) and 2 μL of DNA template. The PCR included a denaturation step at 95°C for 5 min. followed by 45 cycles of 95°C for 3 sec., 60°C for 30 sec., followed by a melting curve with 95°C for 60 sec., 40°C for 60 sec., 65°C for 1 sec. and 97°C for 1 sec. Each sample was run in triplicates along with negative controls in each run. Obtained signal was compared to an in-house E. coli based standard to infer counts of 16S rDNA gene copy numbers.

## Protein extraction for analysis of bacterial secretome

Day 2 culture supernatants were centrifuged to precipitate cells and keratin material. Thereafter proteins were precipitated from the supernatant by addition of TCA to a final concentration of 10%. The solution was incubated on ice for 1 hrs, followed by centrifugation at 15000g for 10 min. Pellets were dissolved in lysis buffer; 6M guanidinium hydrochloride, 10 mM tris (2-carboxyethyl)phosphine (TCEP) (Sigma), 40 mM chloroacetamide (CAA), 50 mM HEPES pH 8.5. Protein concentration was estimated using Bradford (Bio-Rad). For each sample 50 μg protein was diluted 1:10 in 10% acetonitrile (ACN), 50mM HEPES pH 8.5. Trypsin (Pierce[TM] Trypsin Protease, MS Grade, ThermoFisher Scientific) was added (1:100 of trypsin to protein). Samples were incubated overnight at 37°C with shaking at 1000 rpm in a thermo-mixer. Digestion was stopped by adding 10% triflouroacetic acid (TFA) to a final concentration of 2% TFA and an approximate pH of 2–2.5.

## Purification of the digested peptide

Trypsin-digested peptides were purified using a Stage tip protocol as described by Rappsilber [55]. In summary, 3 C18 filters were gently punched out with the help of the sampling tool syringe. Filters were placed at the tip of a 200 μL pipette tip with a plunger. Filters were activated with 30 μL methanol by centrifugation at 1000g for 2 min., followed by 30 μL 100% ACN, and finally 2x 30 μL of 3% ACN with 1% TFA. Care was taken not to dry the disk at any point. Digested peptide samples were loaded onto the filter unit by centrifugation at 1000g until all sample had passed the filter. Bound peptides were washed 2 times using 30 μL of 0.1% formic acid (FA). Peptides were eluted using 2x 30 μL 60% ACN in 0.1% FA. Liquid was evaporated and peptides were re-dissolved in 2% ACN with 1% TFA. Peptide concentration in the samples was estimated with NanoDrop, and 1 μg peptide was loaded for analysis on a Q-Exactive (Thermo Scientific, Bremen, Germany).

## Mass spectrometry

The samples were analysed by liquid chromatography tandem mass spectrometry (LC-MS/MS) and data were recorded in a data-dependent manner, automatically switching between MS and MS/MS acquisition, on a Q-Exactive (Thermo Scientific, Bremen, Germany). An EASY nLC-1000 liquid chromatography system (Thermo Scientific, Odense, Denmark) was coupled to the mass spectrometer through an EASY spray source and peptide separation was performed on a 15 cm EASY-spray columns (Thermo Scientific) with a 2 μm size C18 particles and the inner diameter of 75 μm. The mobile phase consisted of solvents A (0.1% FA) and B (80% ACN in 0.1% FA). The initial concentration of solvent B was 6%, and hereafter gradients were applied to reach the following concentrations: 14% B in 18.5 min, 25% B in 19 min, 38% B in 11.5 min, 60%B in 10 min, 95% B in 3 min and 95% B for 7 min. The total length of the gradient was 70 min. The full scans were acquired in the Orbitrap with a resolution of 120000 and a maximum injection time of 50 ms was applied. For the full scans, the range was adjusted to 350–1500 m/z. The top ten most abundant ions from the full scan were sequentially selected for fragmentation with an isolation window of 1.6 m/z [56], and excluded from re-selection for a 60 sec. time period. For the MS/MS scans, the resolution was adjusted to 120000 and maximum injection time of 80 ms. Ions were fragmented in a higher-energy collision dissociation cell with normalized collision energy of 32% and analyzed in the Orbitrap.

## Mass spectrometry data analysis

The acquired raw data was analyzed using MaxQuant version 1.5.5.1 [57] with the inbuilt Andromeda search engine [58]. Mass tolerance was set to 4.5 ppm (parent ions) and 20 ppm (fragment ions); a maximum of 2 missed tryptic cleavages was permitted. Methionine oxidation and protein N-terminal acetylation were selected as variable modifications and carbamidomethylation of cysteines was set as a fixed modification. A minimum length of seven amino acids per peptide was required. A target-decoy search approach with the default MaxQuant setting of 1% false discovery rate (FDR) was applied for identification at both peptide and protein levels [57]. Normalization was performed with the label-free quantification (LFQ) algorithm [59] in MaxQuant using a required LFQ minimum ratio count of two. Quantification required a minimum ratio count of two, allowing quantification only on unique and razor peptides. The match between runs function was applied to enhance protein identification. All eight biological replicates (replicates A-F, X and Y) were included for protein identification and label-free quantification in MaxQuant. For the following data analysis some replicates were removed due to low data quality; for instance, the biological replicates X and Y had very

low pairwise similarity to the other 6 biological replicates and were therefore excluded from quantification analysis, see S2–S7 Figs.

## Reference genomes

Reference genomes for analysis of the mass spectrometry data were prepared using genomic data of the same strains from prior studies. In short, full genome sequencing had been performed for each isolate and the resulting contigs were annotated with the RAST database [60,61]. Raw reference genomes (contigs) are available online; PRJEB18431 (*X. retroflexus*), PRJEB15265 (*M. oxydans*), PRJEB15263 (*S. rhizophila*), PRJEB15262 (*P. amylolyticus*). The *X. retroflexus* reference genome used for the mass spectrometry data analysis was trimmed to remove peptide sequences shared with any of the other species, as described in a previous study [62]. Identified proteins from the secretome profile are hence identified in a species unique manner. The protein sequences of the RAST annotated *X. retroflexus* reference genome has been appended to the mass spectrometry data upload, see the Data availability section, and is available as part of supplemental materials. Reference genomes were further mapped with protease functions using the BLAST MEROPS function (Release 11.0) [32,63]. Signal peptides were identified in amino acid sequences using Signal-P 4.1 [64].

## Graphs and statistical analysis

Statistical analysis was performed in the R environment (R Development Core Team, 2016). Data visualisation was performed in the R-environment using *ggplot2* [66]. P-values without correction for multiple hypothesis testing are displayed as "p". Corrected p-values are referred to as "$p_{adj}$". Linear regression with post-hoc Tukey pairwise comparison hypothesis testing and single-step p-value correction was applied to make pairwise statistical comparisons. Statistical difference was displayed by dissimilar letters signifying $p_{adj} < 0.05$ and the test type is referred to by "Lin.1". Comparisons of slope means and variance were made by the *lsmeans* package [67] in the R environment, applying a linear regression model including the interaction between day and culture type, with hypothesis testing of the slope with Tukey comparisons and sidak p-value correction, referred to by "Lin.2". Fold changes were compared by a linear regression model with a fixed offset = 1, referred to by "Lin.3". An independent t-test was applied to infer statistical difference between measured and theoretical data on keratin degradation.

Significant changes in protein abundances between mono- and co-cultures of *X. retroflexus* by paired two-sample t-test corrected for multiple hypothesis testing by FDR testing ($q < 0.05$). Principal component analysis (PCA) was performed on log2 transformed protein intensities, using zero centering and unit variance scaling. The analysis was performed with the *Prcomp* functions in the stats R-package [68].

## Results

### Keratin degradation from mono-species cultures

Keratin degradation was tested in shake-flask cultures with keratin liquid medium (KLM) containing 10 mg/mL keratin. After 4 days of cultivation, keratin degradation was evaluated by measuring residual dry-weight. Population growth was evaluated by counting colony forming units (CFU). *X. retroflexus* degraded a significantly larger amount of keratin (2.1 ±0.2 mg/mL, standard deviation [SD]) ($p_{adj} < 0.05$, Lin.1) (Fig 1A) and reached significantly higher CFU counts, as compared to the other single species (S8A Fig). Keratin degradation was observed from *S. rhizophila*, but *M. oxydans* and *P. amylolyticus* displayed limited capability to remove

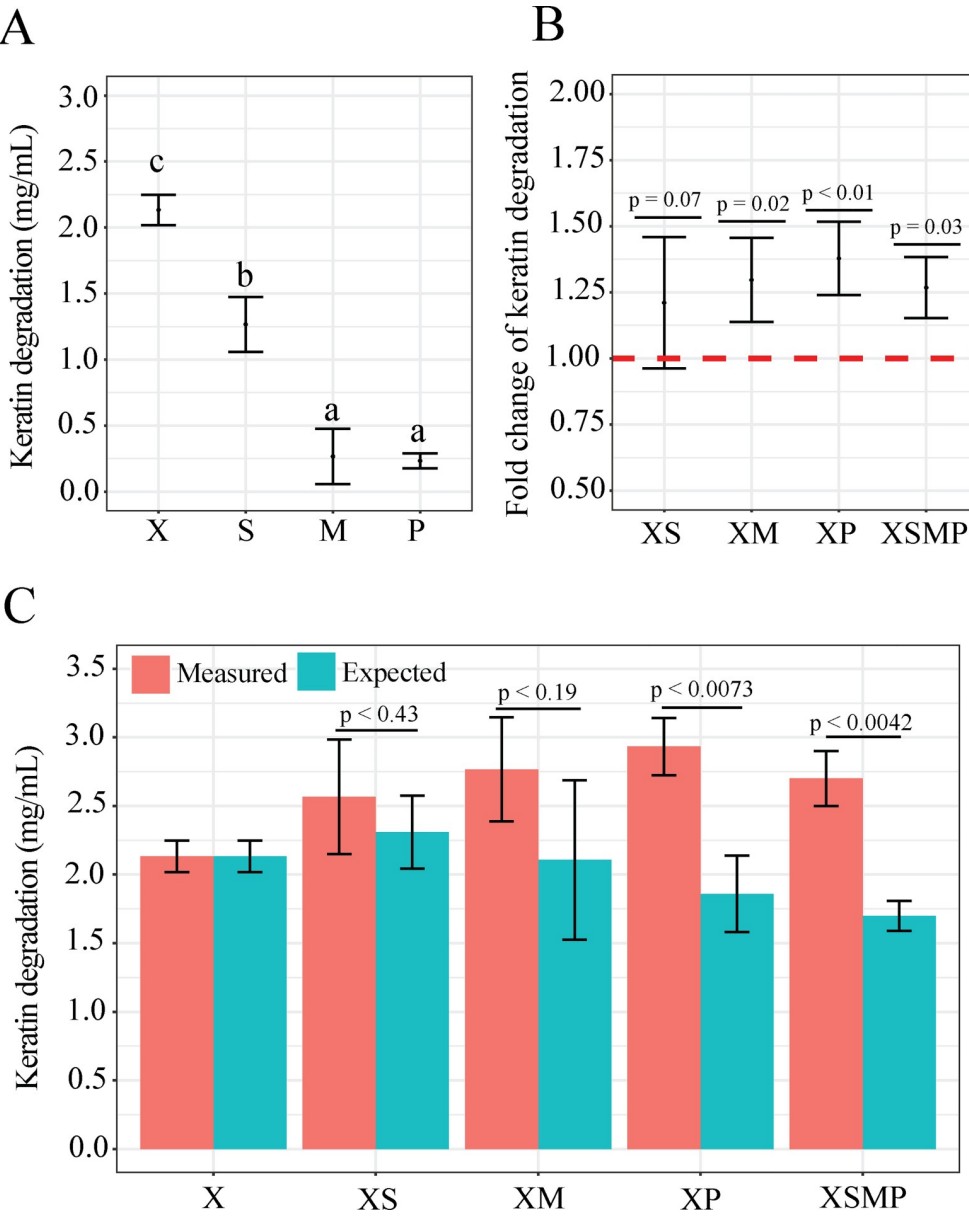

**Fig 1. Keratin degradation by mono- and co-cultures.** *S. rhizophila*, *X. retroflexus*, *M. oxydans* and *P. amylolyticus* are represented by the letters S, X, M and P, respectively. Co-cultures are represented by letters signifying its single species constituents, e.g. XS represents the co-culture of *X. retroflexus* and *S. rhizophila*. Error bars represent standard deviation of three biological replicates. A) Keratin degradation (mg/mL) for mono-species cultures after 4 days of incubation. Different letters define statistical grouping (ascending order, $p_{adj} < 0.05$, Lin.1) B) Fold increase of keratin degradation in co-cultures of *X. retroflexus*, compared to the *X. retroflexus* mono-culture (indicated by dotted red line). Statistical difference is inferred by Lin.3. C) Measured and theoretical keratin degradation (mg/mL) in the *X. retroflexus* mono-culture and co-cultures. 'Measured' refers to the measured keratin degradation, whereas ´expected´ refers to the expected theoretical amount of keratin degraded by the given co-culture. The expected amount was calculated as follows: Expected XS co-culture degradation = Degradation potential of *X. retroflexus* in mono-culture per CFU * CFU count of *X. retroflexus* in XS co-culture + degradation potential of *S. rhizophila* in mono-culture per CFU * CFU count of *S. rhizophila* in the XS co-cultures. For convenience a complete calculation of the XS co-culture has been added as S12 Fig. Significant difference was inferred by a two-tailed independent two-sample t-test.

keratin from the medium. CFU counts in cultures of *S. rhizophila* and *M. oxydans* were higher than those of *P. amylolyticus*, which displayed the significantly lowest CFU counts ($p_{adj} < 0.05$,

Lin.1) (S8A Fig). Co-cultivations increased counts of *P. amylolyticus* (S8B Fig). Keratin degradation was normalized to CFU counts to investigate the degradation potential of each species. With CFU normalization, *X. retroflexus* and *S. rhizophila* were equally efficient in keratin degradation and had a significantly higher degradation than *M. oxydans* (S9 Fig) ($p_{adj}$ < 0.05, Lin.1). Calculation of keratin degradation per CFU for *P. amylolyticus* was omitted, as *P. amylolyticus* did not grow as mono-culture. Cell free supernatants of *X. retroflexus* cultured on keratin was not able to facilitate growth of the other strains, indicating that the environment is very low in nutrients, and that degraded keratin gets metabolised quickly. Degradation of keratin was also supported by an observed alkalization of the media over time for *X. retroflexus* mono- and co-cultures. An alkalization was observed for these cultures from the initial pH of 7 towards pH 8.5.

## Keratin degradation in co-cultures

Degradation from co-cultivation was evaluated for all combinations of dual-species and the 4-species cultures. However, only co-cultures containing *X. retroflexus* had enhanced keratin degradation as compared to the best single species degrader of the individual co-cultures. *X. retroflexus* constituted the majority of these communities according to CFU counts (S8A Fig). The enhanced degradative effect of co-cultures had an average fold-change of 1.29, indicating that keratin degradation in co-cultures was on average enhanced by ~30% as compared to *X. retroflexus* mono-cultures (Fig 1B). Co-cultures XM (mean fold change [MFC] = 1.297, p = 0.02, Lin.3), XP (MFC = 1.38, p < 0.01, Lin.3) and XSMP (MFC = 1.27, p = 0.03, Lin.3) displayed significantly enhanced keratin degradation (Fig 1B). Measurements of keratin degradation for all *X. retroflexus* mono and co-cultures are presented in S10A Fig. Although average total cell counts of co-cultures were not significantly different from that of *X. retroflexus* mono-cultures, keratin degradation was normalized to cell counts to account for potential variations. Measurements of keratin degradation normalised to CFU counts for all *X. retroflexus* mono and co-cultures are shown in S10B Fig. With CFU normalisation, the co-culture of XP and the four-species co-culture still had significantly enhanced keratin degradation compared to that in *X. retroflexus* mono-cultures (S10C Fig).

## Community-intrinsic properties

To test for the presence of community-intrinsic properties, the measured degradation of the individual co-cultures was compared to the predicted maximum degradation from the sum of their constituent single species cultures (S11A Fig). Data included three biological replicates, yielding three measures of community degradation with associated measures of single species degradation. Notably, co-culture XP had a significantly higher mean degradation than the predicted maximum. The XM co-culture non-statistically supported the trend of co-cultures having higher degradation than the predicted level. However, comparing co-culture degradation to the sum of multiple single species cultures can make identification of community-intrinsic properties difficult. Instead co-culture degradation was compared to a normalised theoretical degradation of the co-culture, e.g. for the XS culture (Fig 1C); Expected theoretical degradation of the XS co-culture = Degradation potential of *X. retroflexus* in mono-culture per CFU * CFU count of *X. retroflexus* in XS co-culture + degradation potential of *S. rhizophila* in mono-culture per CFU * CFU count of *S. rhizophila* in the XS co-culture. For convenience a complete calculation of the XS co-culture has been added as S12 Fig. All co-cultures had a higher mean of measured keratin degradation, as compared to the theoretical degradation, indicating that community-intrinsic properties resulted in enhanced degradation. For co-cultures of XP and XSMP (p < 0.0073 and 0.0042 respectively, independent two-tailed t-test) the measured mean

was significantly higher than the theoretical estimate. Normalizing with CFU revealed that co-cultures XP and the four-species community were still significantly more productive than the expected theoretical degradation (S11B Fig).

## Quantitative PCR analysis for keratin adhered cells

Previous studies on microbial degradation of recalcitrant material have linked the degree of degradation to the amount of microbes physically associated to the material. Quantitative analysis using universal eubacterial primers targeting the 16S rRNA gene (qPCR) was used to address the number of cells adhered to the keratin particles, with the assumption that total bacterial cell counts on the particles could indicate a level of biofilm formation on the particles. QPCR analysis was performed on mono- and co-cultures after 2 and 4 days of incubation. Among the mono-cultures; cultures of *X. retroflexus* contained significantly higher counts of copy numbers, as compared to any other type of mono-culture (S13A Fig, p < 0.001, Lin.1) at both time points. *S. rhizophila* displayed the second highest level of copy numbers, although it was not significantly different from *M. oxydans* and *P. amylolyticus*, (S13A Fig).

After two days of incubation only the co-culture XM contained higher amounts of attached cells, as compared to the *X. retroflexus* mono-culture (Fig 2). However, after 4 days of incubation all co-cultures contained more attached cells than the *X. retroflexus* mono-culture, suggesting that a synergistic biofilm production or attachment was at play during co-cultivation. Co-cultures of XM (MFC = 1.61, p = 0.018, Lin.3) and XP (MFC = 1.52, p = 0.014, Lin.3) displayed significantly higher numbers of attached cells, as compared to *X. retroflexus* mono-cultures. Co-cultures of XS and the four-species culture also presented higher fold change, although not significantly (Fig 2). Biofilm counts from co-cultures are included in S13B Fig.

To further investigate the relevance of cell attachment in relation to keratin degradation, we performed a correlation analysis between measured keratin degradation, total cell counts in the supernatant and biofilm counts. The correlation analysis was performed on day 4 data of *X. retroflexus* mono- and co-cultures. Using Pearson's correlation, a significant ($p_{adj}$ = 0.04, FDR corrected p-value) strong positive ($r$ = 0.98) correlation coefficient was observed between keratin degradation and biofilm counts. Oppositely, total CFU counts from the supernatant (as presented in S8 Fig) showed only a non-significant weak negative correlation with keratin degradation. Hence, biofilm formation on the keratin particles is a better predictor for keratin degradation than total CFU counts (S15 Fig and S2 Table). To further support this observation a Spearman's ranked correlation approach was also adopted. Similar to the Pearson's approach a strong positive correlation ($r_s$ = 1) was observed for keratin degradation and biofilm counts, although the correlation in this case was only nominal significant (p = 0.01667, $p_{adj}$ = 0.14 with FDR correction). Again, total culture CFU and keratin degradation only showed a weak negative non-significant correlation (S15 Fig and S3 Table).

## Enzymatic assays and protein measurements

Protease and keratinase activity were measured on a spectrophotometer using azocasein and azokeratin dyed substrates, respectively. Among mono-cultures, protease activity, keratinase activity and soluble protein concentration in the medium were found to increase with time for *X. retroflexus* and *S. rhizophila* (S16A–S16C Fig). For *M. oxydans* and *P. amylolyticus*, no clear proteinase and keratinase activity was observed, and protein content did not change over time. At day 4, the protease and keratinase activity of *X. retroflexus* was significantly higher than that of all other mono-cultures (protease: 39.42 ±2.34 U/mL, keratinase: 34.58 ±12.1 U/mL,

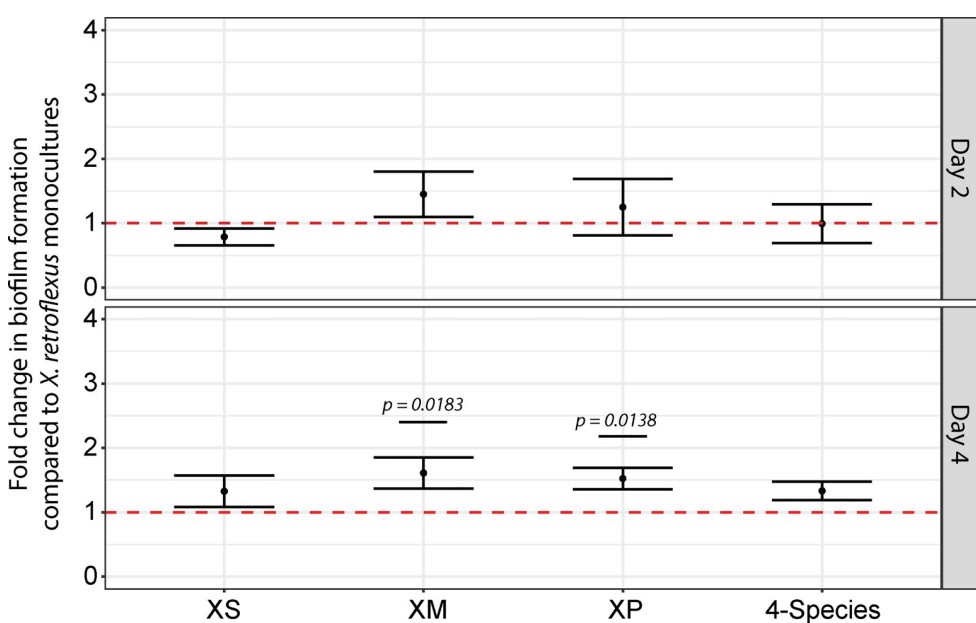

**Fig 2. Quantification of cells associated to keratin particles.** Q-PCR analysis was based on universal eubacterial 16s rDNA gene primers. Keratin particles were isolated from mono- and co-cultures after 2 and 4 days of incubation. Counts of 16S gene copy numbers were believed to correspond to cells associated to the particles in a biofilm state. Co-culture counts were compared to counts of *X. retroflexus* mono-cultures. Dotted red line corresponds to the *X. retroflexus* mono-culture. Co-cultures are represented by letters of their constituent species; e.g. *X.retroflexus–S. rhizophila* (XS), *X.retroflexus–M.oxydans* (XM) and *X.retroflexus–P.amylolyticus* (XP). At day 2, the level of adhered cells was not significantly different between co-cultures and the *X. retroflexus* mono-culture. At day 4, the co-cultures trended an increased fold change of adhered cells, as compared to the *X. retroflexus* mono-culture. Co-cultures of *X. retroflexus–M.oxydans* and *X.retroflexus–P.amylolyticus* had a significantly increased fold change (Lin. 3).

standard deviation, Lin.1). *S. rhizophila* had the second highest proteinase and keratinase activity at day 4 (S16A and S16B Fig), which was significantly higher than *M. oxydans* and *P. amylolyticus* in regards to protease activity, but only nominal significant in regards to keratinase activity (protease: 12.26 ±1.96 U/mL, keratinase: 15.57 ±4.59 U/mL, standard deviation, Lin.1). *X. retroflexus* and *S. rhizophila* (0.29 ±0.01 mg/ml and 0.24 ±0.005 mg/ml respectively, SD) produced significantly higher amounts of soluble protein (Lin.1) compared to *M. oxydans* and *P. amylolyticus*, but there was no significant difference between the production from *X. retroflexus* and *S. rhizophila* (S16C Fig). At day 4, observed protease and keratinase activities for co-cultures of *X. retroflexus* were similar to that of *X. retroflexus* mono-cultures; Only the protease activity of the XS co-culture deviated significantly, and was significantly lower ($p_{adj}$ = 0.0118, Lin.1) (Fig 3A and 3B).

When comparing changes in activity over time, steeper activity increases were observed for some co-cultures, as judged from linear slope's coefficients. Slope coefficients and p-values have been appended as S4 Table. The slope coefficient for the four-species community was higher for both protease and keratinase activity (slope = 7.87 and p = 0.007 and $p_{adj}$ = 0.0276 for protease activity; slope = 8.03 and p = 0.0184 and $p_{adj}$ = 0.0718 for keratinase activity, Lin.2) as compared to the *X. retroflexus* mono-culture, indicating an increased turn-over within the tested time-span. Co-cultures of XM and XP also trended a higher slope coefficient, although not significantly different (p = 0.0571 and $p_{adj}$ = 0.2095 for XM; p = 0.064 and $p_{adj}$ = 0.2323 for XP, Lin.2).

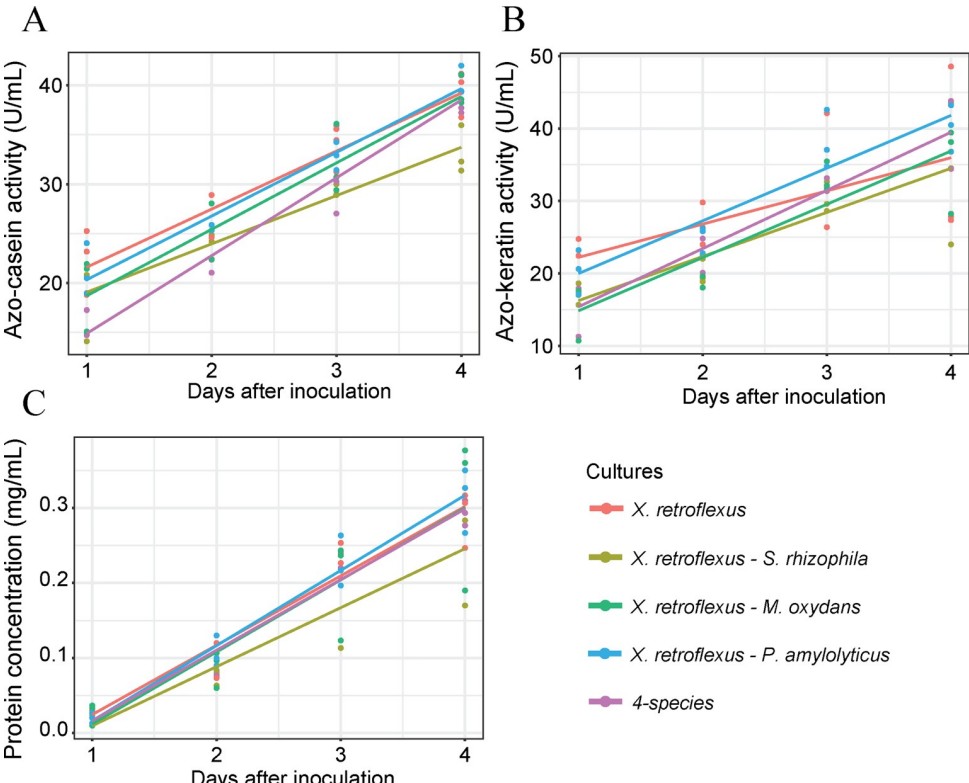

**Fig 3. Enzyme and protein production by the different co-cultures during growth.** Lines represent a linear regression of three biological replicates across sampling time. A) Protease production by different co-cultures using azo-casein as substrate. One unit protease activity was defined (U) as the amount of protein that increased the absorbance by 0.01. B) Keratinase production by different co-cultures using azo-keratin as substrate. One unit keratinase activity was defined as the amount of protein that increases the absorbance by 0.01 under given conditions. C) Protein production from degraded keratin by different co-cultures. Bradford assay was used for protein quantification with BSA as standard.

Although slope coefficients of most co-cultures with *X. retroflexus* were higher, the soluble protein concentration did not vary significantly between these co-cultures and that of *X. retroflexus* mono-cultures (Fig 3C).

Observation of protease activity, keratinase activity and protein concentration was included in the correlation analysis with keratin degradation, total CFU counts and biofilm counts, with day 4 observations. For both the Pearson's and Spearman's ranked correlation analysis keratinase activity displayed a higher correlation coefficient with keratin degradation than protease activity. However, none of the correlations were significant (S14 and S15 Figs and S2 and S3 Tables). Of the three, protein concentration had the second highest correlation coefficients to keratin degradation, although not significant for neither Pearson's nor Spearman's ranked correlations.

## Secretome analysis by mass spectroscopy

To further resolve the increased keratin degradation in co-culture, the secretome of *X. retroflexus* was assessed through LC-MS/MS-based proteomics profiling. Secretome profiling was restricted to *X. retroflexus* as it was; i) the species with the highest potential for degradation and ii) constituted the majority of cells in the co-cultures making it unlikely to obtain good proteome coverage from the other species present in the co-cultures. Secretome profiling was

conducted after two days of cultivation, as preliminary tests showed that secretome profiling quality was inversely proportional to incubation time due to build-up of keratin degradation products that hampered identification of secreted bacterial proteins over time. Identified proteins from *X. retroflexus* were mapped with; i) function and subsystem features from the RAST database [60,61], ii) prediction of signal peptides for secretion to the extracellular environment using SignalP [64] and iii) MEROPS protease functions [32,63]. Data quality is summarised in S2–S7 Figs, and the materials and methods section. Identified proteins were filtered according to the presence of signal peptides based on SignalP [64], narrowing the search towards secreted and/or membrane exported proteins. Total counts of identified proteins after SignalP filtration are summarised in Fig 4. Noticeably, several proteins were uniquely observed in the co-culture sample and not in the mono-culture *X. retroflexus* samples.

Secretome differences of *X. retroflexus* in mono- and co-cultures were addressed by both a supervised and un-supervised statistical approach. A supervised approach was used to make pairwise comparisons of the secretome profiles of the *X. retroflexus* mono-cultures to *X. retroflexus* in the different co-cultures using paired t-test with FDR correction. An unsupervised approach (principal component analysis) was used to identify the proteins causing separation between the different culture types.

For the paired t-test the tested proteins were required to be present in four out of the six biological replicates in both of the compared groups. Fig 4 summarises counts and culture overlap of proteins included in the pairwise comparison. Pairwise comparison identified several proteins of *X. retroflexus* with a significantly changed abundance between the mono-culture and either the four-species or the XS co-culture (Table 1). Notably, many of the proteins with significant changes in abundance were hypothetical proteins without known functions in the RAST or MEROPS database. When comparing the mono-culture to any of the two co-cultures, the same serine protease (S08A) was found to be significantly more abundant in the mono-cultures. An additional predicted protease was significantly more abundant in the mono-cultures when compared to the four-species co-culture (S08A / U69). Oppositely, a glutathionine peroxidase family protein was significantly more abundant for both co-cultures as compared to the mono-culture. From the four-species co-cultures, a nikel transport family protein (NikM) and a PQQ-dependent oxidoreductase, gdhB family protein were also found to be significantly increased.

Proteins with a nominal significantly increased abundance were observed from comparison of the XM and XP co-cultures to the *X. retroflexus* mono-cultures, but no proteins presented a significantly changed abundance after FDR correction. Among these nominal significant proteins, the predicted S08A / U69 protease and a M28B protease were more abundant for the *X. retroflexus* mono-cultures when compared to the XP co-cultures, supporting that *X. retroflexus* alone had increased abundance of secreted proteases. For the XP co-culture, the glutathionine peroxidase family protein was also found to be nominal significantly more abundant than in the mono-cultures.

Principal component analysis did not yield a complete group separation on the two main axes (PCA1: 47.69% and PCA2: 35.97%). However, a trend was observed with co-cultures of XM, XP and the four-species separating away from the XS co-culture and *X. retroflexus* mono-cultures (S17 Fig). Focusing on the top ten features causing group separation in either direction of PCA1 and PCA2, revealed a similar protein profile as presented by the supervised approach (S18 Fig). For instance, the three proteins causing the largest separation on PCA2 were proteases, including two serine proteases and a metallo-protease, supporting that a large difference between the *X. retroflexus* mono-culture and the majority of the co-cultures was related to the abundance of secreted proteases. The protein with the strongest impact on group separation on both PCA1 and PCA2 was a chitinase. However, the effect from the chitinase

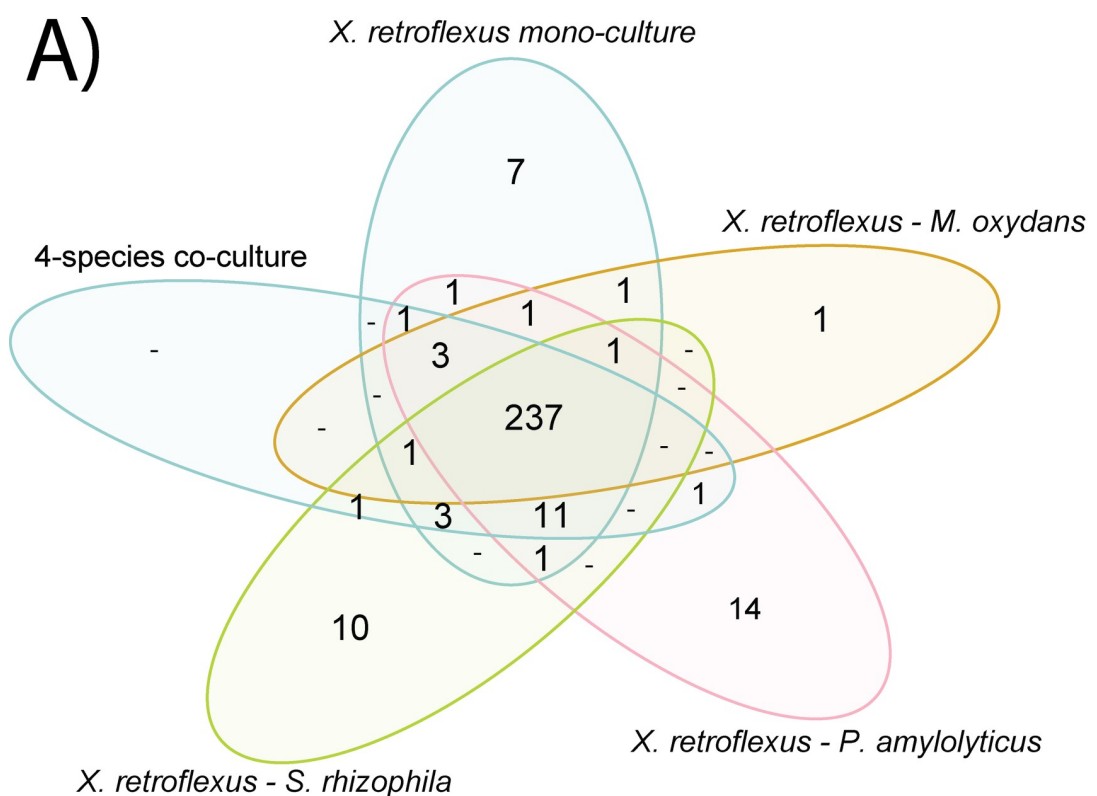

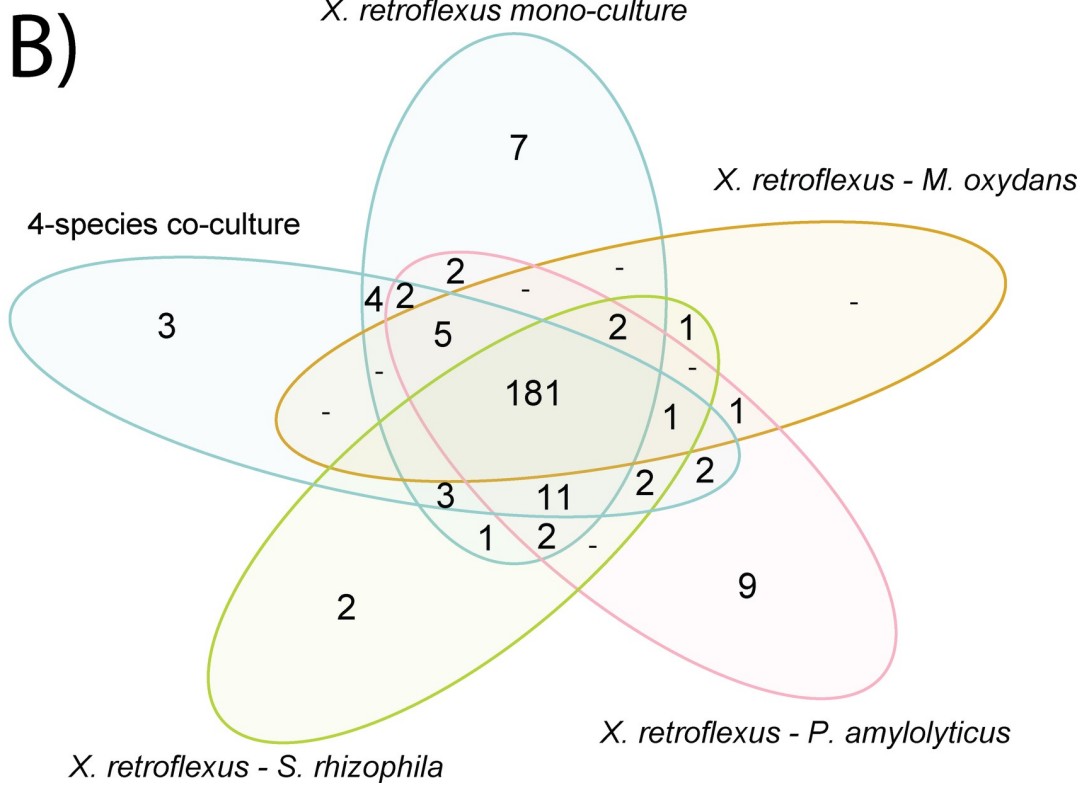

**Fig 4. Number of identified and quantifiable proteins from the secretome LC-MS/MS analysis.** Identified proteins from the secretome samples were filtered by; i) removing the two outlying biological replicates and ii) removing proteins which did not contain a signal peptide. A) Identified proteins and their overlap between different sample groups, only requiring that the protein is observed in one sample of all biological replicates. B) Quantifiable proteins are defined as proteins which were detectable in 4 out of the 6 biological replicates of a given culture. From each co-culture type, only proteins shared with the *X. retroflexus* mono-culture were included in the differential abundance analysis.

was mostly due to its fluctuating presence and absence between culture groups, and not its abundance-variation between groups, which was revealed by performing a binary principal component analysis. Oppositely, the effect of the proteases on PCA2 could not be attributed to mere presence and absence between groups.

## Discussion

*X. retroflexus* and *S. rhizophila* had the highest keratin degradation potential per CFU of the four tested species. However, *X. retroflexus* was overall the species with the highest growth potential from keratin, as it was able to reach higher cell densities during cultivation. Increased keratin degradation was observed for different *X. retroflexus* co-cultures. These co-cultures displayed higher keratin degradation than the theoretical level deduced from mono-culture observations, indicating that enhanced degradation can be a community-intrinsic property. The enhanced degradation was also observed after normalising to the CFU of the co-cultures,

**Table 1. Secreted proteins with significant changes in abundance between mono- and co-cultures after FDR correction.** Significant proteins (after FDR correction, $p_{adj} < 0.05$) were only observed between mono- and co-cultures of *X.retroflexus-S.rhizophila* and the four-species culture. Proteins are listed according to their genome reference name (Protein ID) with the log2 fold change between mono- and co-culture and the FDR corrected p-value of the paired two-sided t-test. Proteins are mapped with RAST subsystem function (Function) and MEROPS functions (MEROPS). [a]The first part of the Protein ID's (fig|305959.5) were removed from the presented IDs.

| Increased in | Co-culture with | Protein ID[a] | Log2(Foldchange) | $P_{adj}$ | MEROPS | Function |
|---|---|---|---|---|---|---|
| Mono-culture | *S. rhizophila* | peg.831 | 0.67 | 0.049672 | | TonB-dependent receptor |
| | | peg.2115 | 0.66 | 0.049672 | S08A | putative autotransporter protein |
| | | peg.670 | 0.42 | 0.049672 | | FIG01200701: possible membrane protein |
| | | peg.1200 | 0.33 | 0.038812 | | Hypothetical protein |
| | | peg.2920 | 0.31 | 0.038812 | | Lipoprotein, putative |
| | 4-species | peg.2115 | 0.92 | 0.022606 | S08A | putative autotransporter protein |
| | | peg.1440 | 0.85 | 0.039562 | S08A | Serine protease, subtilase family |
| | | peg.938 | 0.57 | 0.044323 | | FIG01110996: hypothetical protein |
| | | peg.1499 | 0.52 | 0.023009 | | Flagellar hook-associated protein FlgK |
| | | peg.2920 | 0.48 | 0.044323 | | Lipoprotein, putative |
| | | peg.86 | 0.43 | 0.044323 | | Hypothetical protein |
| | | peg.3823 | 0.40 | 0.040111 | | Hypothetical protein |
| | | peg.2355 | 0.36 | 0.023009 | | Copper metallochaperone, bacterial analog of Cox17 protein |
| | | peg.249 | 0.31 | 0.044323 | | Extracellular ribonuclease precursor (EC 3.1.-.-) |
| Co-culture | *S. rhizophila* | peg.439 | 0.47 | 0.038812 | | Glutathione peroxidase family protein |
| | 4-species | peg.2223 | 0.87 | 0.031398 | | Nikel transport family protein NikM |
| | | peg.2873 | 0.85 | 0.022606 | | Hypothetical protein |
| | | peg.3592 | 0.59 | 0.031398 | | Hypothetical protein |
| | | peg.3855 | 0.56 | 0.044323 | | PQQ-dependent oxidoreductase, gdhB family |
| | | peg.950 | 0.55 | 0.040111 | | Hypothetical protein |
| | | peg.439 | 0.54 | 0.022606 | | Glutathione peroxidase family protein |
| | | peg.3854 | 0.53 | 0.044323 | | Hypothetical protein |
| | | peg.1702 | 0.39 | 0.047914 | | Hypothetical protein |

indicating that in these co-cultures, the individual cells had a higher potential for degrading keratin. Whether in mono- or co-culture, *X. retroflexus* is believed to be the main driver of the degradation. However, as seen from measurements of keratin degradation in relation to protease and keratinase activity, cell association to keratin particles or secretome profiles, all species co-cultured with *X. retroflexus* can affect the culture dynamics to some extent and in turn the keratin degradation. Although the effect of the individual co-cultured species are scattered across different mechanisms and effect sizes, they are all likely to contribute individually and in concert to the community-intrinsic property of enhanced keratin degradation.

Besides enhanced keratin degradation, some co-cultures also displayed enhanced cell association to the keratin particles, potentially in the form of enhanced biofilm formation. Enhanced biofilm formation is a well-known community-intrinsic property, which has been associated to this particular community previously when cultured on e.g. tryptic soy broth [6,50,69,70]. Notably, co-cultures did not yield higher CFU numbers, as compared to the mono-species cultures of *X. retroflexus*, indicating that the enhanced degradation does not lead to further growth. Instead, additional nutrients obtained from the enhanced co-culture degradation could cover the cost of increased attachment to the keratin particles. e.g. enhanced biofilm formation, as matrix production is a costly process. The mechanisms behind the community-intrinsic property then seems to promote an altered keratin turn-over rate or degradation process in the co-cultures, as indicated from the enzymatic activities and the proteomics profiling.

To elucidate the mechanisms behind the community-intrinsic property of enhanced keratin degradation and biofilm formation, the secretome profiles of *X. retroflexus* were analyzed. In general, co-cultures of *X. retroflexus* had a faster turnover rate of keratin. This suggested that the increased keratin degradation observed in co-cultures with *X. retroflexus* could originate from altered expression of specific proteases and/or keratinases. Hence, the enhanced degradation rates could be facilitated by e.g.; i) enhanced protease production by *X. retroflexus* in co-cultures, ii) combined protease secretion from several producers into a communal pool (e.g. S08 proteases), or iii) division of labor between members, with different members producing different proteases which can each facilitate distinct parts of the keratin degradation process. Surprisingly, the secretome profiling of *X. retroflexus* in co-cultures revealed important differences compared to its mono-culture, with a higher production of specific proteases in the mono-culture. For example, the comparison of abundance profiles revealed that the S08 serine protease from *X. retroflexus* was reduced in co-cultures with *S. rhizophila* and *M. oxydans* and in the four-species co-cultures. Proteases from the S08 serine family are key for keratinolityc activity [32], hence variation in their abundance will easily affect keratin degradation. Results indicate that lower abundances of these particular proteases are needed when *X. retroflexus* is present in co-cultures, pointing toward a change in gene expression strategy. As all four species possess genes coding proteases in the S08 family, it may be speculated that the keratinolytic activity by *X. retroflexus* is stimulated by proteases produced by the other strains, thus representing public good resources in co-cultures. Another compatible explanation is that less protease is required to facilitate efficient keratin degradation in co-cultures due to other community-intrinsic properties.

Indeed, keratinolysis is a complex process based on the concerted actions of two categories of enzymes: sulfitolytic and proteolytic enzymes [71,72]. During keratin sulfitolysis, disulphide bonds between polypeptide chains are cleaved, releasing thiol groups and losening the overall structure, making it more accesible for subsequent protease activity [72–74]. Several studies have highlighted that the protease effect is markedly enhanced by disulphide reducing agents including; e.g. i) production of disulphide reductases [74], ii) release of sulphite and thiosulphate [75,76] or iii) a cell-bound redox system [16,30]. Notably, *X. retroflexus* co-cultures with

*S. rhizophila*, *P. amylolyticus* and the four-species increased abundance of proteins from the glutathione peroxidase family, which could be key to balancing the redox environment under the fluctuating culture conditions, due to the production of various reducing agents. This observation hints that sulfitolysis becomes a more favoured strategy for *X. retroflexus* during co-cultivation. In support of a challenged redox environment, superoxide dismutases were among the top-ten most influential proteins on both components 1 and 2 in the unsupervised analysis. Superoxide dismutases also rely on nickel as a co-factor for optimal functioning and, likewise, the nikel transport family protein NikM was more abundant in the four-species culture. Some studies have reported production and/or release of the reducing agent sulphite from *Bacillus* cultures during keratin degradation [75,77], and have linked this to enhanced keratin degradation [75]. In eukaryotes, presence of sulphite is also known to lead to increased lipoperoxidation and disabling of cellular stress defence mechanisms by depleting the glutathione pool and lowering catalase and glutathione peroxidase activities [78]. Providing sulphite has the same effect on prokaryotic cells, it could explain why an increased abundance of e.g. glutathione peroxidase is observed from co-cultures.

Another potential mechanism enabling enhanced degradation in co-cultures could be expression of silent pathways or genes. Several proteins were found to be uniquely present in the co-culture setting, hinting towards the importance of silent pathways or genes (Fig 4). However, care should be taken when observing uniquely present proteins, as these proteins might only be unique due to the detection limit of the mass spectrometry setup. Further optimization of the experimental setup might enable further verification of the uniqueness of these unique proteins, and enable a verification of the importance of silent pathways or genes, which by example could cause a changed proteolytic profile. Unravelling proteolytic profile of all mono-species and all co-culture combinations thereof, might enable a better understanding of each species' contribution to the keratin degradation, and maybe also support the relevance of unique proteins. Zimogram gel analysis with inclusion of different inhibitors, e.g. PMSF, EDTA, IAA, N-ethylmaleimide and pepstatin, could be included in future attempts to investigate and unravel proteolytic profiles of single- and co-cultures.

A final mechanism contributing to enhanced keratin degradation from co-culture could be the previous documented enhanced biofilm formation observed from mixed species communities, such as the investigated community [6]. It has been speculated that efficient keratin degradation is associated with physical contact between the degrading cells and the keratin [79,80], similar to what has been observed from bacterial degradation of other types of recalcitrant material [81]. In our case the strong positive correlation between biofilm and keratin degradation supports the relevance of cell association in regards to keratin degradation. Functionally, close association of cells with the keratin could enable a continuous supply of reductants to break disulfide bridges [75,79], thereby facilitating degradation. Combining proteome and biofilm observations could indicate that enhanced degradation from co-cultures occurred from successful synergistic biofilm establishment on the keratin particle surface, resulting in more efficient sulfitolysis, freeing peptide chains for further proteases activity to occur, leading ultimately to a more efficient strategy for keratin degradation.

## Conclusion

Interspecific interactions between microbes may result in synergistic effects, were the outcome for a given trait exceeds the mere sum of individual species contributions in separation. These so-called "community-intrinsic properties" could be very advantageous in an industrial setting aiming to degrade e.g. recalcitrant material. These community-intrinsic properties could lead

to a larger substrate turnover by e.g. activating silent pathways or shifting the degradation regime.

Although the level of keratin degradation observed from this community is not on par with that of other known degraders, the study verify that higher degradation rates of recalcitrant keratinous material could be achieved using mixed species communities. Our results provide a framework for potential future applied bioprospecting of relevant microbial community-intrinsic properties.

## Supporting information

**S1 Table. Cell inoculation in keratin liquid media for keratin degradation.** For each single species cultures of 100 mL keratin liquid keratin was inoculated with 1 mL of an $OD_{600nm}$ = 0.7 adjusted culture. (a) CFU per mL at OD 0.7 for each single species. (b) CFU $^*$ $mg^{-1}$ keratin with inoculation of 1 mL of a given single species culture at $OD_{600nm}$ = 0.7.
(DOCX)

**S2 Table. Pearson's correlation coefficients and associated p-values and FDR adjusted p-values ($p_{adj}$) for all variables include in S14 Fig.** Protease.Act refers to measured protease activity, Keratinase.Act refers to measured keratinase activity, Protein.Conc refers to measured protein concentration in the culture supernatant, Total_CFU refers to the summed CFU counts for all species in the culture, KeratinLoss refers to amount of keratin removed in the culture during cultivation, and Biofilm refers to counts of 16S rDNA gene copies.
(DOCX)

**S3 Table. Spearman's ranked correlations coefficients and associated p-values and FDR adjusted p-values ($p_{adj}$) for all variables include in S15 Fig.** Protease.Act refers to measured protease activity, Keratinase.Act refers to measured keratinase activity, Protein.Conc refers to measured protein concentration in the culture supernatant, Total_CFU refers to the summed CFU counts for all species in the culture, KeratinLoss refers to amount of keratin removed in the culture during cultivation, and Biofilm refers to counts of 16S rDNA gene copies.
(DOCX)

**S4 Table. Mean slope coefficients and p-values for the liner model on azo-casein and azo-keratin degradation.** *S. rhizophila*, *X. retroflexus*, *M. oxydans* and *P. amylolyticus* are represented by the letters S, X, M and P, respectively. Co-cultures are represented by letters signifying it single species constituents, e.g. XS represents the co-culture of *X. retroflexus* and *S. rhizophila*.
(DOCX)

**S1 Fig. CFU $^*$ $mL^{-1}$ for each single species culture.** Over night cultures in TSB were diluted and plate spread on TSA for a set of fixed OD values. Each plating at a given OD was performed in duplicates. Solid lines corresponds to a linear regression across the data points, with the light grey area showing the rolling average of the 95% confidence interval across the data points. Single letters corresponds to the individual single species accordingly; X is *Xanthomonas retroflexus*, S is *Stenotrophomonas rhizophila*, M is *Microbacterium oxydans* and P is *Paenibacillus amylolyticus*.
(DOCX)

**S2 Fig. Principal component analysis of biological replicates included in the protein identification from secretome profiling.** Identified proteins were filtered for the presence of signal peptides by SignalP, only including proteins which contained signal peptides. Principal component analysis was performed on Log2 transformed protein intensities using zero centering

and unit variance scaling for the PCA analysis with the prcomb R-package. Biological replicates X and Y were clearly differentiating from the other biological replicates, which would hamper proper protein quantification. These two replicates were therefore excluded prior to further analysis.
(DOCX)

**S3 Fig. Multi-scatter plot of biological replicates from the secretome profiling of *X. retroflexus* mono-cultures.** Numbers represents Pearson's correlation between individual samples. The biological replicates X and Y had very low correlation scores with the other replicates, which could influence protein quantification.
(DOCX)

**S4 Fig. Multi-scatter plot of biological replicates from the secretome profiling of *X. retroflexus*–*S. rhizophila* cultures.** Numbers represents Pearson's correlation between individual samples. The biological replicates X and Y had very low correlation scores with the other replicates, which could influence protein quantification.
(DOCX)

**S5 Fig. Multi-scatter plot of biological replicates from the secretome profiling of *X. retroflexus*–*M. oxydans* cultures.** Numbers represents Pearson's correlation between individual samples. The biological replicates X and Y had very low correlation scores with the other replicates, which could influence protein quantification. Similarly, the biological replicate B also represented very low correlation scores, indicating problems with either sample prep or mass spectrometry analysis.
(DOCX)

**S6 Fig. Multi-scatter plot of biological replicates from the secretome profiling of *X. retroflexus*–*P. amylolyticus* cultures.** Numbers represents Pearson's correlation between individual samples. The biological replicates X and Y had very low correlation scores with the other replicates, which could influence protein quantification.
(DOCX)

**S7 Fig. Multi-scatter plot of biological replicates from the secretome profiling of the four-species co-cultures.** Numbers represents Pearson's correlation between individual samples. The biological replicates X and Y had very low correlation scores with the other replicates, which could influence protein quantification.
(DOCX)

**S8 Fig. Counts of colony forming units across culture types.** a) CFU counts of mono- and co-cultures. *S. rhizophila*, *X. retroflexus*, *M. oxydans* and *P. amylolyticus* are represented by the S, X, M and P, respectively. Co-cultures are represented by letter combinations of its single species constituents, e.g. XS represents the co-culture of *X. retroflexus* and *S. rhizophila*. Bars of mono-cultures represent the mean of three biological replicates with error bars showing standard deviation of the replicates. Stacked bars of co-cultures are the summed average of each species from three biological replicates with error bars displaying the standard deviation of the summed mean of all species in the co-culture. Statistical difference by a linear regression with post-hoc Tukey's HSD pairwise hypothesis testing and single-step p-value correction (Lin.1). The red dotted line signifies separates mono and co-cultures. Mono-cultures were statistically compared to each other. Co-cultures were only statistically compared to *X. retroflexus* mono-culture. Statistical difference was found between some mono-cultures, as signified by dissimilar lettering (ascending order, $p_{adj} < 0.05$). No statistical difference was found between the summed averages of co-cultures and the average of the *X. retroflexus* mono-culture. b)

CFU counts of *P. amylolyticus* as mono and co-cultures. Statistical difference was found between cultures, as signified by dissimilar lettering ($p_{adj} < 0.05$, Lin.1).
(DOCX)

**S9 Fig. Keratin degradation per CFU for *X. retroflexus*, *S. rhizophila* and *M. oxydans* as mono-cultures.** *S. rhizophila*, *X. retroflexus*, *M. oxydans* are represented by the S, X, and M respectively. Keratin degradation is calculated as pico-gram keratin degraded per CFU from the cultures. Point represents the mean of three biological replicates with error bars displaying standard deviation. Statistical significance was inferred by linear regression with post-hoc Tukey's HSD pairwise hypothesis testing and single-step multiple step correction, as signified by dissimilar lettering ($p_{adj} < 0.05$, Lin.1).
(DOCX)

**S10 Fig. Keratin degradation with and without CFU correction for Co-cultures of *X. retroflexus*.** *S. rhizophila*, *X. retroflexus*, *M. oxydans* and *P. amylolyticus* are represented by the S, X, M and P, respectively. Co-cultures are represented by letter combinations of its single species constituents, e.g XS represents the co-culture of *X. retroflexus* and *S. rhizophila*. a) Keratin degradation by *X. retroflexus* mono and co-cultures. Mean of keratin degradation from three biological replicates, with error bars showing standard deviation. Statistical difference was inferred by a pair-wise comparison of co-culture to mono-culture by a linear regression p-value corrected by single-step method. Both nominal and adjusted p-values are displayed for tests having a nominal significant p-value. Means of co-cultures were as follows; *X. retroflexus-S. rhizophila* (2.6 ±0.42 mg/mL, std.dev), *X. retroflexus-M. oxydans* (2.8 ±0.38 mg/mL, std. dev), *X. retroflexus-P. amylolyticus* (2.9 ±0.21 mg/mL, std.dev) and four-species community (XSMP) (2.7 ±0.12 mg/mL, std.dev). Both nominal and adjusted p-values are displayed for tests having a nominal significant p-value. **b)** Keratin degradation per CFU by *X. retroflexus* mono and co-cultures. Keratin degraded per CFU was calculated as the total amount of measured keratin degraded in the culture, divided by the total count of CFU from the culture. Mean of keratin degradation from three biological replicates, with error bars showing standard deviation. Statistical difference was inferred by a linear regression model (Lin.1). c) Fold-change in keratin degradation per CFU by co-cultures of *X. retroflexus*, related to the *X. retroflexus* mono-culture (indicated by dotted red line). Statistical difference was inferred by Lin.3.
(DOCX)

**S11 Fig. Measured and expected theoretical keratin degradation from *X. retroflexus* mono and co-cultures, with and without correction for species composition and CFU counts.** *S. rhizophila*, *X. retroflexus*, *M. oxydans* and *P. amylolyticus* are represented by the letters S, X, M and P, respectively. Co-cultures are represented by letters signifying it single species constituents, e.g. XS represents the co-culture of *X. retroflexus* and *S. rhizophila*. Error bars represent standard deviation of three biological replicates. Significant difference was inferred by independent two-sample t-test. 'Measured' refers to the experimentally measured keratin degradation. a) Comparison of measured degradation, for mono and co-cultures of *X. retroflexus*, to the theoretical amount of potential keratin degradation. The theoretical value refers to the sum of keratin degraded by each of the individual single-species cultures constituting the co-culture. b) Keratin degradation per CFU by mono- and co- cultures of *X. retroflexus*. Expected value refers to the theoretical amount of keratin degraded by co-cultures calculated as follows; the theoretical amount of keratin to be degraded by a given co-culture was estimated as the sum of keratin, which could be degraded by the amount of CFU from each species observed in the given culture. The amount degraded by each species was inferred from the potential of the respective mono-species cultures. The summed keratin degradation was then normalised

against the total CFU from the respective co-cultures.
(DOCX)

**S12 Fig. Calculation of expected theoretical keratin degradation for the *X. retroflexus—S. rhizophila* co-culture from their individual single species degradation profiles.**
(DOCX)

**S13 Fig. Copy numbers from Q-PCR analysis, based on universal eubacterial 16s rDNA primers.** Performed on keratin particles isolated from mono- and co-cultures at both 2 and 4 days of incubation. Copy numbers are believed to correspond to cells associated to the particles in a potential biofilm state. Mono- and co-cultures are represented by letters of their respective species; e.g. *X. retroflexus* (X), *S. rhizophila* (S), *M. oxydans* (M) *and P. amylolyticus* (P) for mono-cultures and e.g. *X. retroflexus–M. oxydans* (XM) for co-cultures. Statistical difference was inferred by Lin.1, as signified by dissimilar lettering ($p_{adj} < 0.05$). a) Counts from mono-cultures after 2 and 4 days of incubation. *X. retroflexus* produced significantly higher numbers than any of the other mono-cultures. b) Counts after 2 and 4 days of incubation from co-cultures. No significant difference was observed between counts from cultures after 2 and 4 days of incubation. After 4 days of incubation all co-cultures trended a higher level of counts than the *X. retroflexus* mono-culture.
(DOCX)

**S14 Fig. Correlation matrix with Pearson's correlations between investigated variables.** Averaged values of all biological replicates across each culture type after 4 days of cultivation was used for the correlation analysis. Protease.Act refers to measured protease activity, Keratinase.Act refers to measured keratinase activity, Protein.Conc refers to measured protein concentration in the culture supernatant, Total_CFU refers to the summed CFU counts for all species in the culture, KeratinLoss refers to amount of keratin removed in the culture during cultivation, and Biofilm refers to counts of 16S rDNA gene copies. Color intensity and circle size corresponds to the size of the correlation coefficient ranging between -1 and 1. The larger and darker blue the circle is displayed the closer the correlation coefficient is t 1. The larger and darker red the circle is displayed, the closer the correlation coefficient is to -1. Stars within the circles correspond to level of significance for the FDR corrected p-values: no star refers to $P_{adj} > 0.05$, * refers to $0.05 > P_{adj} < 0.01$, ** refers to $0.01 > P_{adj} < 0.001$ and *** refers to $0.001 > P_{adj}$.
(DOCX)

**S15 Fig. Correlation matrix with Spearman's ranked correlations between investigated variables.** Averaged values of all biological replicates across each culture type after 4 days of cultivation was used for the correlation analysis. Protease.Act refers to measured protease activity, Keratinase.Act refers to measured keratinase activity, Protein.Conc refers to measured protein concentration in the culture supernatant, Total_CFU refers to the summed CFU counts for all species in the culture, KeratinLoss refers to amount of keratin removed in the culture during cultivation, and Biofilm refers to counts of 16S rDNA gene copies. Color intensity and circle size corresponds to the size of the correlation coefficient ranging between -1 and 1. The larger and darker blue the circle is displayed the closer the correlation coefficient is t 1. The larger and darker red the circle is displayed, the closer the correlation coefficient is to -1. Stars within the circles correspond to level of significance for the FDR corrected p-values: no star refers to $P_{adj} > 0.05$, * refers to $0.05 > P_{adj} < 0.01$, ** refers to $0.01 > P_{adj} < 0.001$ and *** refers to $0.001 > P_{adj}$.
(DOCX)

**S16 Fig. Enzyme and protein production during growth by the mono-species cultures.**
Three biological replicates with lines representing linear regressions. a) Protease production
by different combinations using azo-casein as substrate. One unit (U) protease activity was
defined as the amount of protein that increased the absorbance by 0.01 under given condi-
tions. b) Keratinase production by different combinations using azo-keratin as substrate. One
unit keratinase activity was defined as the amount of protein that increased the absorbance by
0.01 under given conditions. c) Protein production from degraded keratin by different combi-
nations. Bradford assay was used for protein quantification with BSA as standard. The higher
initial protein concentration in *M. oxydans* and *P. amylolyticus* (also found in the blank
media) reflects the presence of other proteins apart from keratin in the solution, which may be
due to use of crude keratin. Since *M. oxydans* and *P. amylolyticus* do neither utilize nor
degrade protein from the supernatant, they have the initial and almost same protein level dur-
ing the entire incubation period.
(DOCX)

**S17 Fig. Principal component analysis of culture types from the secretome analysis of *X.
retroflexus*.** Data was filtered to remove the two outlying biological replicates followed by a
removal of one outlying technical replicate of the *X. retroflexus–M. oxydans* and *X. retroflexus–
S. rhizophila* culture, respectively, where the sample analysis on the mass spectrometer had not
yielded data of sufficient quality. Identified proteins were filtered for the presence of signal
peptides by SignalP, only including proteins which contained signal peptides. Principal com-
ponent analysis was performed on Log2 transformed protein intensities using zero centering
and unit variance scaling for the PCA analysis with the prcomb R-package. Although some
group overlap was observed, the co-cultures of *X. retroflexus–M. oxydans* and the four-species
culture separated somewhat from the cultures of *X. retroflexus* and *X. retroflexus–S. rhizophila*
on PCA2.
(DOCX)

**S18 Fig. Influence of top ten variables from the top two loadings from the principal com-
ponent analysis.** Data was filtered to remove the two outlying biological replicates followed by
a removal of one outlying technical replicate of the *X. retroflexus–M. oxydans* and *X. retro-
flexus–S. rhizophila* culture, respectively, where the sample analysis on the mass spectrometer
had not yielded data of sufficient quality. Identified proteins were filtered for the presence of
signal peptides by SignalP, only including proteins which contained signal peptides. Principal
component analysis was performed on Log2 transformed protein intensities using zero center-
ing and unit variance scaling for the PCA analysis with the prcomb R-package. The top ten
protein variables having the largest effect on PCA1 and PCA2 in both positive and negative
direction were extracted and mapped with MEROPS and RAST pathway function. Proteins
without labels were hypothetical proteins from the RAST database without any known MER-
OPS function. PCA1 is most strongly influenced by the presence of a Chitinase (Fig|305959.5.
peg.995). The effect of the Chitinase could mostly be explained by its variation in presence and
absence between groups. PCA2 was strongly influenced by the variating abundance of three
proteases. Two S08A serine proteases (fig|305959.5.peg.2700 and fig|305959.5.peg.3465) and a
M72 metallo-endopeptidase (fig|305959.5.peg.2077)
(DOCX)

## Acknowledgments

The authors would like to acknowledge Anette Hørdum Løth for her assistance running labo-
ratory setups and Jakob Russel for his assistance with data management in the R environment.

The authors also thanks Erwin Schoof and Lene Holberg Blicher for their assistance with mass spectrometry analysis at DTU Proteomics Core, Technical University of Denmark. Keratin substrate for experimental trials was generously supplied by Daka, SARIA group, Denmark.

## Author Contributions

**Conceptualization:** Poonam Nasipuri, Jakob Herschend, Samuel Jacquiod, Søren J. Sørensen.

**Data curation:** Poonam Nasipuri, Jakob Herschend, Asker D. Brejnrod.

**Formal analysis:** Poonam Nasipuri, Jakob Herschend, Asker D. Brejnrod, Samuel Jacquiod.

**Funding acquisition:** Søren J. Sørensen.

**Investigation:** Poonam Nasipuri, Jakob Herschend.

**Methodology:** Poonam Nasipuri, Jakob Herschend, Samuel Jacquiod.

**Project administration:** Samuel Jacquiod, Søren J. Sørensen.

**Resources:** Roall Espersen, Birte Svensson, Mette Burmølle.

**Software:** Asker D. Brejnrod.

**Supervision:** Jakob Herschend, Mette Burmølle, Samuel Jacquiod, Søren J. Sørensen.

**Validation:** Jakob Herschend, Jonas S. Madsen, Samuel Jacquiod, Søren J. Sørensen.

**Visualization:** Poonam Nasipuri, Jakob Herschend, Asker D. Brejnrod.

**Writing – original draft:** Poonam Nasipuri, Jakob Herschend, Samuel Jacquiod.

**Writing – review & editing:** Poonam Nasipuri, Jakob Herschend, Asker D. Brejnrod, Jonas S. Madsen, Roall Espersen, Birte Svensson, Mette Burmølle, Samuel Jacquiod, Søren J. Sørensen.

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
