## [Decision Letter · Decision Letter 0]

31 Oct 2019

PONE-D-19-26241

Community-intrinsic properties enhance keratin degradation from bacterial consortia

PLOS ONE

Dear Dr. Sørensen,

Thank you for submitting your manuscript to PLOS ONE. After careful consideration, we feel that it has merit but does not fully meet PLOS ONE’s publication criteria as it currently stands. Therefore, we invite you to submit a revised version of the manuscript that addresses the points raised during the review process.

We would appreciate receiving your revised manuscript by Dec 15 2019 11:59PM. To enhance the reproducibility of your results, we recommend that if applicable you deposit your laboratory protocols in protocols.io, where a protocol can be assigned its own identifier (DOI) such that it can be cited independently in the future. For instructions see: http://journals.plos.org/plosone/s/submission-guidelines#loc-laboratory-protocols

We look forward to receiving your revised manuscript.

Kind regards,

Arun K. Bhunia

Academic Editor

PLOS ONE

Journal Requirements:

Reviewers' comments:

Reviewer's Responses to Questions

**Comments to the Author**

1. Is the manuscript technically sound, and do the data support the conclusions?

Reviewer #1: Yes

Reviewer #2: Yes

2. Has the statistical analysis been performed appropriately and rigorously? 

Reviewer #1: Yes

Reviewer #2: Yes

3. Have the authors made all data underlying the findings in their manuscript fully available?

Reviewer #1: Yes

Reviewer #2: Yes

4. Is the manuscript presented in an intelligible fashion and written in standard English?

Reviewer #1: No

Reviewer #2: Yes

5. Review Comments to the Author

Reviewer #1: The authors presented a study that evaluates the degradation of keratin by 4 different bacteria and the co-cultivation of these species. Interesting results are presented here, however some corrections and complements to experiments I understand can be done to improve the quality of the work.

Lines 120-121 (pg 5). Replace “The four bacterial strains used in this study were previously isolated from soil and characterized” for “The four bacterial strains used in this study were previously isolated from soil and identified by morphological analysis and 16S rRNA gene sequences.”

Line 126 (pg 6). “Liquid cultures were...” What was the growth conditions (for "X" hrs at ºC and rpm)?

Line 129 (pg 6). “1%, 0.5% and 0.25% inoculum...” Please, provide the inoculum size in relative biomass (g/L: bacterial biomass/media volume). It is possible by using a standard curve: O.D. x dry biomass. The definition of inocullum is essential to reproduce the experiments!

Line 148 (pg 7) “min at 4°C or min. at 4°C.”

Line 187 (pg 8). To separate keratin there was filtration on filter paper. To separate cells from supernatant, was the material centrifuged?

Line 361 (pg 15) “rhizophila in mono-culture * CFU count of X. retroflexus in co-culture” or “rhizophila in mono-culture * CFU count of S. rhizophilain co-culture”

Lines 358-361 (pg 15) “Expected degradation for XS co-culture = degradation potential of X. retroflexus in mono culture * CFU count of X. retroflexus in co-culture + degradation potential of S. rhizophila in mono-culture * CFU count of X. retroflexus in co-culture.” The figure legend (Fig 1C and S4B) are better that this explanation. Please, provide a better information about this calculation. Use an example.

Line 405 (pg 16) Spectrophotometer? It is a colorimetric method. Please correct for “Protease and keratinase activity were measured via spectrophotometer using azocasein...”

Lines 414-415 (pg 17) “but not significantly higher in regards to keratinase activity...”. It is not true. The keratinolytic activity from S. rhizophila was higher than M. oxydans and P. amylolyticus, in which no enzyme activity was noted.

Lines 420-422 (pg 17) “Most co-cultures had protease and keratinase activities similar to that of X.

retroflexus mono-cultures, when comparing end-point activity at day 4 for mono- and

co-cultures of X. retroflexus.” Improve. It is repetitive!

Line 526 (pg 23) “X. retroflexus had the highest growth potential from keratin, reaching higher cell”. It does not sound well!

Line 531 (pg 23) “ a community-intrinsic property”. In this work, it would be interesting to check the diversity of proteases in mono and co-cultures by zimogram gel analysis and effect of inhibitors (PMSF, EDTA, IAA, N-ethylmaleimide and pepstatin A). The authors justify only analysis of proteases from X. retroflexus because the low protein concentration/bacterial growth of the other species. So, that suggested assays would be useful to evaluate the protease profile in the secretome of each mono and co-culture.

Line 599 (pg 26) Correct “keratin degradtion”

Line 632 (pg 27) “advantageous in an industrial setting”. Provide examples!

Reviewer #2: This manuscript presents an interesting approach on the study of biodegradation mechanisms of recalcitrant keratinous material. The article describes that co-cultures of keratinolytic strains can present synergistic collaboration to facilitate keratin degradation. Although cooperative mechanisms of microbial biodegradation have been described, similar studies on keratin are scarce and this work presents a solid experimental data to support the hypothesis. The manuscript is generally well written but needs revision for some awkward phrasing and typos.

Specific points:

1) Lines 34-35. How can you define the isolates as “well-characterized synergistic” at this point?

2) Line 105-106. Although complete keratin degradation is not usually achieved by individual strains, some isolates can do that. Thus, the sentence needs rewriting.

3) Several abbreviations are not defined in the Methods section. This point should be carefully revised.

4) Line 129. This sentence is confused, please rewrite. In addition, sentences should not be initiated by numbers, please check throughout the manuscript.

5) Line 151. The source of azokeratin should be described.

6) Line 167. This heading is somewhat confusing. In this case, it seems that “DNA extraction from biofilm adhered to keratin particle” is sufficient.

7) Line 192. The source and grade of trypsin should be provided.

8) Results. Figures are not properly placed/labeled in the manuscript, which impairs a clear evaluation.

9) Fig. 1. The values of “expected” keratin degradation in 1C do not match with those observed in 1A. It seems that values correspond to a normalized theoretical degradation but calculation is not clearly described here.

10) Fig. 2. Legend to this figure is somewhat unclear and should be revised. Data from Fig. 2 and Fig. S5 could be correlated with data from Fig. S1.

11) Figure S13 could be included as Fig. 4 in the main body of this manuscript.

12) Lines 479-480 and Table 1. Predictive function for some hypothetical proteins can be assigned by putative conserved domains tool of BLAST algorithm. The protein ID codes provided in Table 1 do not allow searching in EMBL or NCBI databases. Please provide adequate accession numbers or gi.

13) Lines 575-578. This hypothesis should be reinforced correlating the observed values for proteolytic/keratinolytic activities.

14) Line 614. Could some differentially expressed proteins be associated with bacterial adhesion, aggregation or biofilm formation?

15) The manuscript should be thoroughly revised for typing and grammar errors.

6. PLOS authors have the option to publish the peer review history of their article (what does this mean?). If published, this will include your full peer review and any attached files.

Reviewer #1: Yes: Ronivaldo Rodrigues da Silva

Reviewer #2: No

---

## [Author Response · Author response to Decision Letter 0]

17 Dec 2019

We have appended a document to the revised manuscript called "ResponseToReviewers". Please refer to this document for specific comments to editors and reviewers, as some of their comments required us to include data analysis, figures and tables, in order to properly address the comments.

---

## [Decision Letter · Decision Letter 1]

8 Jan 2020

Community-intrinsic properties enhance keratin degradation from bacterial consortia

PONE-D-19-26241R1

Dear Dr. Sørensen,

We are pleased to inform you that your manuscript has been judged scientifically suitable for publication and will be formally accepted for publication once it complies with all outstanding technical requirements.

With kind regards,

Arun K. Bhunia

Academic Editor

PLOS ONE

Additional Editor Comments (optional):

accept

Reviewers' comments:

Reviewer's Responses to Questions

**Comments to the Author**

1. If the authors have adequately addressed your comments raised in a previous round of review and you feel that this manuscript is now acceptable for publication, you may indicate that here to bypass the “Comments to the Author” section, enter your conflict of interest statement in the “Confidential to Editor” section, and submit your "Accept" recommendation.

Reviewer #1: All comments have been addressed

Reviewer #2: All comments have been addressed

2. Is the manuscript technically sound, and do the data support the conclusions?

Reviewer #1: Yes

Reviewer #2: Yes

3. Has the statistical analysis been performed appropriately and rigorously? 

Reviewer #1: Yes

Reviewer #2: Yes

4. Have the authors made all data underlying the findings in their manuscript fully available?

Reviewer #1: Yes

Reviewer #2: Yes

5. Is the manuscript presented in an intelligible fashion and written in standard English?

Reviewer #1: Yes

Reviewer #2: Yes

6. Review Comments to the Author

Reviewer #1: The manuscript is ok. ALL corrections have been addressed. I would like to recommend this manuscript for publication in PLoS One

Reviewer #2: The manuscript contains novel and interesting information on microbial consortia involved in keratin degradation. This article has been improved following the Reviewers' comments and the revised version is recommended for publication.

7. PLOS authors have the option to publish the peer review history of their article (what does this mean?). If published, this will include your full peer review and any attached files.

Reviewer #1: No

Reviewer #2: No

---

## [Editor Report · Acceptance letter]

13 Jan 2020

PONE-D-19-26241R1 

Community-intrinsic properties enhance keratin degradation from bacterial consortia 

Dear Dr. Sørensen:

I am pleased to inform you that your manuscript has been deemed suitable for publication in PLOS ONE. Congratulations! Your manuscript is now with our production department. 

With kind regards,

on behalf of

Dr. Arun K. Bhunia 

Academic Editor

PLOS ONE